# What should be included in case report forms? Development and application of novel methods to inform surgical study design: a mixed methods case study in parastomal hernia prevention

Charlotte Murkin ,[1] Leila Rooshenas,[1] Neil Smart,[2] I R Daniels,[2] Tom Pinkney,[3] Jamshed Shabbir,[4] Timothy Rockall,[5] Joanne Bennett,[6] Jared Torkington ,[7] Jonathan Randall,[4] H T Brandsma,[8] Barnaby Reeves ,[9] Jane Blazeby ,[1] Natalie S Blencowe ,[1] On behalf of the CIPHER Study

For numbered affiliations see end of article.

**Correspondence to**
Natalie S Blencowe;
natalie.blencowe@bristol.ac.uk

## ABSTRACT

**Objectives** To describe the development and application of methods to optimise the design of case report forms (CRFs) for clinical studies evaluating surgical procedures, illustrated with an example of abdominal stoma formation.
**Design** (1) Literature reviews, to identify reported variations in surgical components of stoma formation, were supplemented by (2) intraoperative qualitative research (observations, videos and interviews), to identify unreported variations used in practice to generate (3) a 'long list' of items, which were rationalised using (4) consensus methods, providing a pragmatic list of CRF items to be captured in the Cohort study to Investigate the Prevention of parastomal HERnias (CIPHER) study.
**Setting** Two secondary care surgical centres in England.
**Participants** Patients undergoing stoma formation, surgeons undertaking stoma formation and stoma nurses.
**Outcome measures** Successful identification of key CRF items to be captured in the CIPHER study.
**Results** 59 data items relating to stoma formation were identified and categorised within six themes: (1) surgical approach to stoma formation; (2) trephine formation; (3) reinforcing the stoma trephine with mesh; (4) use of the stoma as a specimen extraction site; (5) closure of other wounds during the procedure; and (6) spouting the stoma.
**Conclusions** This study used multimodal data collection to understand and capture the technical variations in stoma formation and design bespoke CRFs for a multicentre cohort study. The CIPHER study will use the CRFs to examine associations between the technical variations in stoma formation and risks of developing a parastomal hernia.
**Trial registration number** ISRCTN17573805.

## INTRODUCTION

A major challenge facing surgical research is that surgical procedures are considered to be complex healthcare interventions, comprising multiple components that can act independently or interdependently to

### STRENGTHS AND LIMITATIONS OF THIS STUDY

⇒ This study used a scientific and systematic process to inform the design of case report forms for a main study.
⇒ Supplementing literature reviews with qualitative methods helped to highlight important surgical technique variations that may otherwise not have been identified.
⇒ A potential limitation is that literature searches were undertaken up to 2016 and therefore further important studies (generating more themes) may have been missed.
⇒ Due to the in-depth and multimodal data collection, this study was time consuming so future studies are needed to streamline the process.

influence outcomes.[1] This complexity means that even surgical procedures labelled with the same name are frequently undertaken in different ways. Such variation has implications for the design and conduct of studies evaluating surgical interventions. It can introduce bias into randomised controlled trials (RCTs) due to difficulties in establishing exactly how the interventions in each group were delivered. In non-randomised studies, a lack of consideration for variations in surgical technique may compromise the full exploration of relationships between exposures and outcomes and result in misidentification of important factors, leading to criticism of the results.[1] This may be particularly problematic in studies that aim to identify risk factors for the 'failure' of a procedure (ie, recurrence of the same condition, or development of a postoperative complication).

One way to understand the heterogeneity of surgical procedures (within randomised and non-randomised studies) is to record accurately how their constituent components are performed within case report forms (CRFs). This makes it possible to document variation in delivery of an intervention, and subsequently link this to outcomes, enabling study findings to be contextualised within routine practice. It is therefore crucial that consideration is given to how details of intervention delivery are recorded in CRFs. While it may be impractical to monitor all components of an intervention, it is necessary to provide clarity about which details are expected to be documented and very important information for interpreting study results. Currently, however, methods for identifying the components of interest are lacking, and there is a tension between recording all possible surgical components and streamlining data collection. This paper describes the development of novel methods to optimise the documentation of intervention components in CRFs for studies evaluating surgical procedures.

## METHODS

The research described in this paper was undertaken as preliminary work prior to the UK Cohort study to Investigate the Prevention of parastomal HERnia (CIPHER study), a multicentre cohort study (HTA-14/166/01) aiming to investigate the technical risk factors for developing parastomal hernia (PSH). Technical risk factors can be defined as variations in operative technique that may predispose patients to a PSH. PSH is a common complication following stoma formation surgery, and although various patient-related risk factors have been proposed, technical factors relating to the initial surgical procedure are much less certain. The preliminary work comprised three phases: (1) identification of all of the possible technical factors that may contribute to PSH formation, using literature reviews and mixed qualitative research methods to deconstruct stoma formation into its component parts and steps; (2) amalgamation of these findings into a 'long list' of potential factors influencing PSH development and (3) rationalisation of the 'long list' into a pragmatic but systematically determined list of final data items to be collected in the CIPHER study.

### Phase 1: identifying the technical risk factors for PSH development

Potential risk factors relating to the technical aspects of stoma formation were defined as those that were: (a) 'known' (defined as reported in the literature, identified from systematic literature searches) and (b) 'unreported but practised' (identified using mixed qualitative methods). Methods to identify each type are described in full further.

### Systematic literature reviews

To comprehensively identify 'known' technical risk factors, snowballing methods were used to find papers cited by and citing prominent articles relevant to the research question. Snowballing was selected as an efficient alternative to conventional search strategies. Database searches using terms such as 'stoma' and 'parastomal hernia' yielded many abstracts, of which only a small proportion were relevant. In contrast, snowballing provided a manageable volume of results and in-depth review of full-text papers to identify all relevant information relating to the intricacies of stoma formation.

### Search strategy

Snowballing began using an index paper,[2] defined as a high-quality paper known to contain information relevant to the study question. The index paper[3] was selected by two senior surgeons both with extensive experience of surgical research. Papers cited by the index paper were then screened (backward snowballing) using broad inclusion criteria (ie, written in English and providing technical details of small or large bowel stoma formation). Once backward snowballing was complete, all papers citing the index paper were retrieved (forward snowballing) using a citation tracking service. A further five index papers[4–8] were then selected to continue snowballing. These papers were identified through discussion with experts, with a focus on data-rich, high-quality literature (eg, reviews). At the end of the snowballing process, two newly published RCTs were identified. Because of their potential relevance, it was agreed they should form part of the literature review, but further snowballing was not undertaken.[9 10]

### Data collection and analysis

Searches and analyses of the articles were performed in tandem. Once identified as relevant, papers were uploaded to NVivo V.10 (QSR International, Melbourne, Australia). Included papers were read in full to identify all text regarding the technical variations of stoma formation. Once relevant text was identified, a qualitative content analysis was performed, and the text was coded into themes, arranged according to the component parts of stoma formation.[11] A proportion of texts were double coded by a second reviewer (LR) to iteratively refine the coding framework. Once saturation was felt to be achieved, a further reviewer (NSB) tested the framework by coding further works to identify any additional themes. This was performed twice.

### Mixed qualitative study

To identify the 'unreported but practised' potential risk factors for PSH, an exploratory study was performed using mixed qualitative methods. This comprised non-participant observation[2] and digital video data capture of stoma formation within the operating theatre and semistructured interviews with surgeons and stoma nurses. This work had three objectives: (1) to confirm, refine, refute and develop the 'known and reported' factors identified in the literature work, (2) to uncover and explore 'unknown but practised' technical variations in stoma

formation that may influence PSH development and (3) to triangulate the non-participant observation and semistructured interview to allow for a more comprehensive understanding of stoma formation techniques.[2] For example, semi-stuctured interviews allowed for opportunity to explore and explain the practices observed and tease out practices that informants reported conducting outside of the observed cases; non-participant observation allowed for direct insight into real cases, providing detail that might have been omitted from interviews.

### Sampling

Operations featuring stoma formation were purposefully sampled from two National Health Service hospital centres, with the aim of observing stoma formation performed by different surgeons, rather than the same surgeon creating multiple stomas. The sampling of healthcare professionals was based on a 'key informant' purposeful approach, whereby individuals with exclusive specialist expertise are selected as the most productive sample to speak to and observe.[12] Clinicians with a role in conducting stoma surgery or providing aftercare were therefore approached. This included stoma nurses and surgeons performing stoma surgery either routinely (ie, colorectal surgeons) or as part of the emergency service (ie, upper gastrointestinal surgeons and colorectal surgeons), at both consultant and registrar level. Approximately 32 surgeons (including registrars) were performing stoma formation, and six stoma nurses were employed at the time of data collection between the two trusts. Sampling continued up until the point of saturation, whereby during parallel analysis, no new themes were uncovered, and additional data were not adding any further explanatory or descriptive benefit. The decision about when to cease further sampling/data collection was guided by the core research team's (CM, LR and NSB) assessment of the 'tipping point', whereby further interviews were not deemed unlikely to be of global benefit for fulfilling the study aim and objectives. This was similar to what others have referred to as the point of 'informational redundancy'[13]: a point at which 'new data tend to be redundant of data already collected'.[14] In this case, we reached a point whereby no new items were emerging from concurrent analysis of interviews. While there was a possibility of this changing with further data collection, the team made a judgement to cease, once the benefits of continued sampling were weighed up against the consequences of delaying the next steps of the mixed-methods project.

### Data collection

Non-participant observations of stoma formation were undertaken by CM (a medically qualified researcher) and documented as field notes. Observations were supplemented by digital videos of the procedure, filmed by theatre staff or a member of the medical illustration team using portable video recorders in conjunction with above-head theatre cameras (where available). Recording started when stoma formation began and stopped on completion, when the stoma bag was attached. All recordings were stored on encrypted and secure devices.

Semistructured interviews were conducted by CM using a topic guide. The topic guide was a series of open-ended questions and prompts/probes, arranged into broad topics to guide the interviewer.[11] Informed by literature review findings, the topic guide was pretested in two interviews (not included in the analysis) to test for flow, relevance and whether the questions provoked discussion. Interview topics derived from the category themes identified during the literature review and were updated iteratively as new themes emerged during the interviews. All interviews were digitally audio-recorded and transcribed verbatim. Interviews with surgeons were held directly after an observation wherever possible, to add explanatory weight to the practices observed and explore inconsistencies between what was observed and what was said. Due to time constraints, this was sometimes not possible and instead interviews were held at a mutually convenient time.

### Data analysis

Efforts were made to analyse the data immediately after each interview or observation was completed. This enabled emerging findings to be explored further and scrutinised in the interviews and/or observations that followed. Field notes from observations were aligned with each digital video and watched in full, unedited, by two researchers (CM and NSB) independently. The researchers took notes on the videos by documenting the surgeon's movements to create a stepwise account of the surgery, which was imported into NVivo. Semistructured interviews with surgeons and nurses were transcribed verbatim and also imported into NVivo.

All transcripts were analysed thematically by coding, line by line, to construct themes relating to technical variations in stoma formation. This was an iterative process. CM and LR coded a sample of the transcripts in the early stage of the analysis and met to discuss their coding and agree a consistent framework. LR scrutinised subsequent coded transcripts, and CM, LR and NSB met regularly to discuss data interpretation and evolution of the coding framework.

### Phase 2: creating a long list of potential risk factors for PSH

Findings from the literature review and mixed qualitative methods were amalgamated by comparing and contrasting emerging themes. Two researchers, including one senior general surgeon (NSB), reviewed each theme and compared the findings of the two data collection processes. The wording of themes and organisation into categories was iteratively refined, combining similar themes and retaining distinction where relevant. Some themes were reclassified as non-technical factors, such as preoperative bowel preparation. This was a long and iterative process as the coding framework from both phases interacted and created more complexity within the

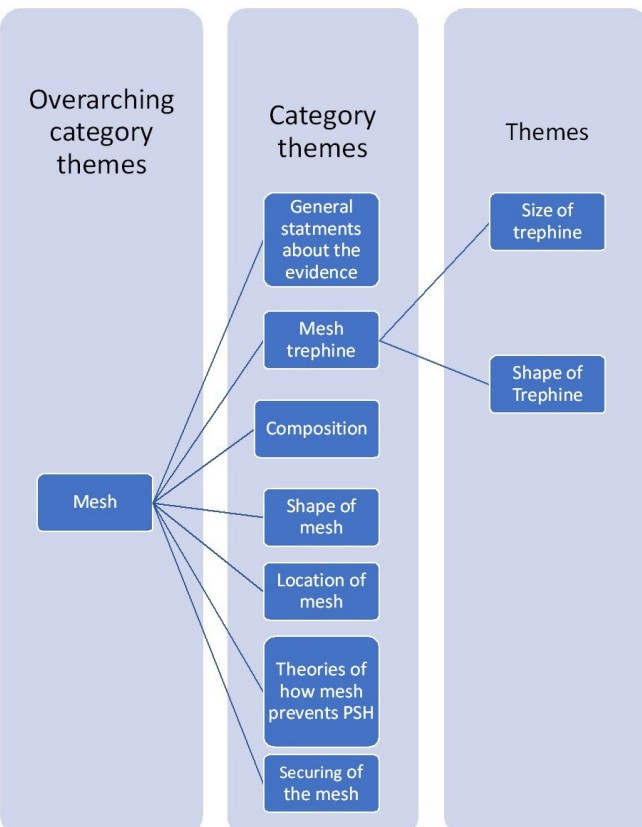

**Figure 1** Example of the coding framework for mesh trephine size. PSH, parastomal hernia.

framework of overarching themes, category themes and themes (figure 1). Themes did not always use the same language and were difficult to untangle for the more complex concepts. For example, many surgeons referred to the 'Modified Sugarbaker' technique as 'Sugarbaker' in the interviews, whereas in the literature, the two were distinct with different meanings. After removing duplicates, refining the description of the themes and creating exhaustive response categories, a 'long list' of discrete risk factors was compiled, organised sequentially according to the steps of stoma formation.

### Phase 3: agreement of the final list of data items to be collected in the CIPHER cohort study

A consensus meeting was held to achieve agreement about the final list of data items to be collected. Participants were selected using a key informant approach, where participants were deliberately selected for their expertise as either appointed colorectal surgeons or for their experience in research and publication in the field of parastomal hernia prevention to represent a broad spread of views. Trial managers were invited to provide views on the practicality of recording information (eg, measurement of adequate bowel mobilisation). Each risk factor in the 'long list' was introduced by the meeting facilitator and discussed in turn. Attendees were encouraged to consider whether each data item was 'essential', 'desirable' or 'not required' for inclusion in the CIPHER

CRFs. Consideration was also given to the wording of each item to ensure there was no ambiguity to the data items and that the response categories were mutually exclusive. When required, researchers (NSB and CM) supplemented discussions with information about the origin of each data item (eg, from qualitative work or reported in the literature). In the case of items reported in the literature, they also described strength of evidence (eg, from a systematic review, RCT or case series).

If consensus (ie, unanimous agreement) was achieved about whether an item was 'essential' or 'not desirable', the facilitator moved onto the next item for discussion. Items where consensus was not met or those considered 'desirable' were carried forward to the next 'round' of discussion, when they were reconsidered in light of how all other items had been prioritised.

Further meetings with the trials design team were held to map the items prioritised and refined through consensus using design principles for CRFs.

### Patient and public involvement (PPI)

PPI work was central to the design and conduct of the CIPHER study. During conception of the project, it was discussed at the Association of Coloproctology of Great Britain and Ireland Patient Consultation Exercise (attended by representatives from national patient support groups), and the study team met separately with patient representatives from colostomy, ileostomy and urostomy organisations. Two patients are members of the Study Steering Committee. Items relating to the seniority of surgeon forming the stoma, and whether they were supervised, were of particular interest to patients attending the PPI meeting, and they agreed strongly that these should be included in the CRFs.

## RESULTS
### Phase 1: identifying the risk factors for PSH development
#### Literature review

Data saturation was achieved after undertaking forward and backward snowballing of six index papers (table 1). A total of 480 references were screened, and of these, 130 articles finally included with a publication date ranging from 1958 to 2016. Within these articles, 138 items relating to the technical aspects of stoma formation, hypothesised to influence PSH prevention, were identified.

#### Mixed qualitative methods

Six procedures involving stoma formation were observed (table 2). The ages of patients whose stoma formation was observed ranged from 31 to 84 years. Thirteen health professionals were interviewed across two centres (table 3). No surgeon that was approached declined an interview. However, it was not possible to interview four willing surgeons due to their conflicting time commitments. Six stoma nurses were eligible for inclusion, three consented and none further were approached. Interviews were conducted by CM who was acquainted with three of

**Table 1** Approach taken to literature snowballing

| Index paper | Description of index paper | References forward snowballing | References backward snowballing | Articles excluded | Articles not located | Duplicates | Articles eligible | Additional articles included in the review (excluding duplicates) |
|---|---|---|---|---|---|---|---|---|
| Shabbir, 2012[3] | Systematic review | 34 | 42 | 12 | 1 | 0 | 63 | 42 |
| Aquina, 2014[4] | Review article | 108 | 13 | 12 | 1 | 15 | 93 | 38 |
| Hauters, 2016[5] | Prospective cohort study | 29 | 15 | 1 | 0 | 19 | 24 | 10 |
| Prudhomme, 2016[6] | Randomised control trial | 30 | 0 | 0 | 0 | 19 | 11 | 6 |
| Hotouras, 2013[7] | Systematic review | 115 | 42 | 12 | 6 | 37 | 102 | 26 |
| Hardt, 2013[8] | Systematic review | 52 | 9 | 4 | 3 | 29 | 25 | 6 |
| Additional articles suggested by experts: Brandsma, 2016[9] and López-Cano, 2016[10] | Randomised control trials | 2 | 0 | 0 | 0 | 0 | 0 | 2 |

the surgeons interviewed. Interviews lasted between 17:22 and 1:13:95 min (mean=27:45 min). There was a broad range of experience across the clinicians interviewed and consensus meeting participants: their length of time in role ranged from <1 to 16 and from <1 to 17 years, respectively.

The qualitative work identified 150 items relating to technical factors that were thought to influence the development of PSH.

### Phase 2: amalgamation of phase 1 findings into a long list of data items

Following amalgamation of the themes for the phase 1 data collection processes and exclusion of duplicates, 180 technical factors were identified (40 were unique to the literature, 63 unique to the qualitative work, 77 appeared in both). This long list of individual data items was arranged into tables to present the data in a format that was easy to manage for the consensus meeting. Formatting of the long list into tables resulted in the splitting and condensing of some data items as well as the addition of further response categories such as 'other' where appropriate. The long list was reviewed by two senior colorectal surgeons from different centres who were not attending the consensus meeting. This was to check for the researchers' understanding of terminology and also to ensure that all factors had adequately been identified.

**Table 2** Characteristics of the non-participant observation patient sample

| Sex | Ethnicity | Type of stoma formed | Surgical approach to stoma formation | Indication for stoma formation | Planned or unplanned surgery | Centre number |
|---|---|---|---|---|---|---|
| Male | White British | End colostomy | Laparoscopic | Bowel management for paraplegia | Planned | 1 |
| Male | White British | End colostomy | Laparoscopic | Bowel management for multiple sclerosis | Planned | 1 |
| Female | White British | End colostomy | Open | Bowel cancer | Planned | 2 |
| Female | White British | End ileostomy | Laparoscopic | Inflammatory bowel disease | Planned | 1 |
| Male | White British | End ileostomy | Converted laparoscopic to open | Bowel cancer | Planned | 2 |
| Male | White British | End colostomy | Open | Sigmoid volvulus | Unplanned | 2 |

**Table 3** Healthcare professional interviewee characteristics

| Sex | Specialty | Grade/role |
| --- | --- | --- |
| Female | Upper gastrointestinal | Consultant |
| Female | Colorectal | Stoma nurse |
| Female | Colorectal | Stoma nurse |
| Male | Colorectal | Consultant |
| Male | Upper gastrointestinal | Consultant |
| Male | Upper gastrointestinal | Consultant |
| Male | Colorectal | Consultant |
| Male | Colorectal | Consultant |
| Male | Colorectal | Consultant |
| Male | Colorectal | Registrar |
| Male | Colorectal | Consultant |
| Female | Colorectal | Stoma nurse |
| Male | Colorectal | Consultant |

This review resulted in the addition of 12 new data items that were felt to be of interest. A total of 207 data items were taken to the consensus meeting for discussion (a segment of the long list is provided in online supplemental file 1, table 1, table 1).

The data items were arranged into eight overarching category themes: (1) surgical approach to stoma formation; (2) trephine formation (skin and subcutaneous tissue, muscle layers, other); (3) reinforcing the stoma trephine with mesh; (4) closure of the lateral space; (5) use of the stoma as a specimen extraction site; (6) stoma snugness; (7) closure of other wounds formed during the procedure; and (8) creating and securing the stoma lumen. Each overarching category theme is discussed in greater detail further using examples from the qualitative work to demonstrate their reported significance to PSH prevention.

### Surgical approach to stoma formation
Findings from both the literature and qualitative work suggested several aspects of the surgical approach to stoma formation may influence the development of PSH. Many variations to operative technique were identified in the literature, with further depth and detail added by the qualitative work. These included: the overall approach to stoma formation (open, laparoscopic and mixed); the section of bowel used to create the stoma; the length of bowel mobilised; whether the stoma site was 'premarked'; and the route of the stoma through the abdominal wall (table 4).

### Trephine formation
The term 'trephine' describes the hole or opening in the abdominal wall for the stoma to pass through. Trephine formation was divided into three subsections: skin and subcutaneous fat, muscle layers and other.

### Skin and subcutaneous tissue
All the variations in this theme were identified in the non-participant observations, with some being common to the literature and/or the semistructured interviews. Various techniques were employed by surgeons to form the trephine of the skin, including the instrument used to create the incision as well as size and shape of the incision (eg, see online supplemental file 1, table 2, extracts 1 and 2). How the trephine was formed through the subcutaneous tissue was also raised as a possible technical factor that could impact on PSH rates. Variations involving the subcutaneous tissue included whether adipose tissue was excised or divided, and (if excised), whether this was done in the shape of a cone or column (see online supplemental file 1, table 2, extracts 3–6).

### Muscle layers
Both the literature and mixed qualitative methods identified multiple variations relating to trephine formation through the abdominal muscle layers. Many were uniquely found in either the literature or the qualitative work. Overall, both the literature and qualitative work indicated that factors surrounding the muscle trephine may be important for PSH prevention. Subthemes within this theme included the location of the muscle trephine, methods used to make the incision through the anterior rectus sheath, posterior rectus sheath, how the stoma trephine was created through the muscle fibres, and methods to measure and dilate the stoma trephine. Interviews revealed different surgical preferences for where the trephine should be created, with some reporting that trephine formation within the rectus sheath was superior for PSH prevention, while others were unsure of the significance of the trephine location. Informants also described diverse practices for how the trephine was formed through the different layers of the abdominal wall. Some described using circular incisions to incise the anterior and posterior sheath. This was thought to prevent PSH through an improved distribution of tensile forces (see online supplemental file 1, table 2, extract 11). Others preferred a cruciate incision to create the trephine with a 'minimal defect' (see online supplemental file 1, table 2, extract 12).

### Trephine formation: other
This overarching category theme related to intraoperative nerve injury and factors relating to laparoscopic procedures, such as level of gas inflation during the trephine formation and use of the stoma trephine as a port site. The subthemes specific to laparoscopic procedures were identified solely in the qualitative work, particularly the semistructured interviews. As shown in the extract further, the level of inflation and use of the stoma site as a port-site were anticipated by some to impact on PSH occurrence:

Extract 1: 'I would generally deflate, I think it's important to deflate the abdomen before making a stoma cut…so from what we've talked about so far, what

**Table 4** Examples from the qualitative data representing the development of the overarching category theme *'Surgical approach to stoma formation'*

| Subtheme | Extract |
|---|---|
| Surgical approach to stoma formation | Extract 1: 'I would favour using a laparoscopic technique if that was technically possible. The reason being that you can actually confirm the anatomy so you can perform an end colostomy, which I think reduces the risk of parastomal formation and the other complications such as prolapse and retraction. You can actually mobilise the colon to bring it up to the abdominal wall'. (HCP: BRI0022, surgeon, lower GI, RDE)<br>Extract 2: 'There is talk, that maybe sometimes laparoscopic surgery they end up with more hernias, but to me I think it's when they've had a full-blown laparotomy and then they have a stoma form. I think they're probably more at risk because they've got double… They appear to have weaker muscles, don't they?'. (HCP: BRI0003, stoma nurse, BRI) |
| The section of bowel used to create the stoma | Extract 3: 'It does, you can do end stoma rather than a loop. The loop ones I think we tend to have a lot more trouble with prolapse, retraction and herniation because you have to make a bigger cut to bring up the loop of the colon'. (HCP: BRI0022, surgeon, lower GI, RDE)<br>Extract 4: 'An end seems less likely; a loop ileostomy seems more likely to get a parastomal hernia than an end ileostomy. I'm trying to think if colostomies seem less likely to get an earlier-on hernia and more likely to get a later-on hernia'. (HCP: BRI0002, stoma nurse, BRI) |
| Length of bowel mobilised | Extract 5: interviewer: 'What do you think about the amount of bowel mobilised? Do you think that would make a difference to parastomal hernias?'.<br>Respondent: 'Probably. Yes, you're probably right, probably more with stoma prolapse or retraction, rather than parastomal hernias. I suppose if you have a prolapsing stoma, it would widen the defect. I don't know'. (HCP: BRI0009, surgeon, upper GI, BRI) |
| Premarked stoma site | Extract 6: 'The stoma site is pre-marked at two sites above and below and left lateral to the umbilicus. Both sites have been tied with a suture'. (Observation CM: BRI0014, end colostomy, laparoscopic, BRI)<br>Extract 8: interviewer: 'You also mentioned preoperative marking of the stoma site. Do you think that will make a difference to the parastomal hernia rates?'.<br>Respondent: 'It probably doesn't, it is probably more about having a better site for the patient in terms of a place where they can change it. I suppose sometimes if it is marked in a position that is not ideal for a surgeon it can be more challenging for us. If anything is more challenging you might increase things marginally, but probably not'. (HCP: BRI0022, surgeon, lower GI, RDE) |
| Route of the stoma through the abdominal wall | Extract 9: 'Technical factors associated with parastomal. So, I suppose one thing to address is whether we do this as a trans-peritoneal, or an extra peritoneal approach. So, years ago, in the '60s and '70ss, and maybe even more recently, it was quite common for the stomas to be tunnelled, pre-peritoneally, laterally, so essentially what you were doing is you'd have the bowel up laterally against the abdominal wall or the under surface, and then it would come out through the muscles as an extraperitoneal stoma. That may have an impact, I don't know, that's never been subjected to a randomised trial comparing it to the trans-peritoneal approach, where the bowel simply just comes through the abdominal wall without tunnelling it. So I think that's a possible surgical technique factor'. (HCP: BRI0004, surgeon, lower GI, BRI) |

you want is a hole that's closely mirrors the diameter of the bowel so not too much so that you can get a parastomal hernia, not too little so that it… you're gonna get it pinched. So, to create that doing it under tension with the CO2 inside pushing up against the fascia just doesn't make sense to me'.

Extract 2: interviewer: 'I've seen people perform laparoscopic surgery through the stoma site…so will use a [Alexis] wound protector and a glove; do you think that will have an affect on parastomal hernia rates?'.

Respondent: 'Yeah I think exactly the same reason so I don't do it. Nice idea I know people are desperate to keep their wounds down, but if you've got a sub total colectomy for a permanent ileostomy which is with the patient for the rest of their life, I think you're gonna get a hernia and they would probably, I have never actually done it with anyone but they

would probably narrow the stoma hole down later and they're using stitches, and yet we've said that repairing a parastomal hernia with stitches has got a 100% recurrence rate'. (HCP: BRI0036, surgeon, lower GI, BRI)

### Reinforcing the stoma trephine with mesh

The use of mesh to reinforce the stoma site at the primary stoma formation was an overarching theme throughout the literature and qualitative work with both sources providing rich data. The use of mesh was a commonly published topic; however, the qualitative work demonstrated uncertainty on its value:

Extract 1: 'I put an intraperitoneal mesh to stop the parastomal hernia. That is my anecdotal experience, I've done 10 or 15 of those and I think they are less likely to get hernias, but even if they get hernias,

they're less likely to be symptomatic'. (HCP: BRI0010, lower GI, BRI)

Extract 2: 'I've seen meshes eroding through the bowel, so I'm reluctant to use it'. (HCP: BRI0010, surgeon, lower GI, BRI)

There were also variations in practice and views in relation to the type of mesh, its size and shape, location, positioning, the size and shape of trephine through the mesh and how the mesh was secured (see online supplemental file 1, table 3).

### Closure of the lateral space

Closure of the lateral spaces was referred to in both the literature and the semistructured interviews as a method of PSH prevention. One surgeon theorised that a method to close the lateral space could influence PSH formation by closing a space where a hernia could form, while others were unsure:

Extract 1: 'I think having a permanent one, potentially closing the space, you know, probably in the short term, you know, before a bit of bowel has a chance to sort of get its foot through the door metaphorically, after you're closing that space off so there's nothing that can sneak around there while you're waiting for a bit of fibrosis to form, makes sense to me'. (HCP: BRI0032, surgeon lower GI, RDE)

Closure of the lateral space was performed through 'fascial fixation', the act of attaching the stoma, either the bowel wall (serosa) or the mesentery (the bowels attachment to the posterior abdominal wall where its blood supply derives from) to the anterior abdominal wall.[15] Fascial fixation was discussed in a portion of the semistructured interviews and was occasionally performed by two surgeons within their clinical practice (extracts 1 and 2 further) but not others. Subthemes within this category included fixation of the mesentery to the abdominal wall and fixation of the stoma to the abdominal wall:

Extract 1: 'The other thing maybe is whether or not you choose to [tack] the stoma, or choose to tack the bowel to the fascia. So, in does that somehow reduce your risk of parastomal hernia formation? Again, I don't know… I would suture the serosal surface. That's another thing; I don't do any tacking of the mesentery to anything, it's just literally bowel wall'. (HCP: BRI0004, surgeon lower GI, BRI)

Extract 2: 'I suture the colon to the rectus and to the skin separately…' (HCP: BRI0009, surgeon, upper GI, BRI)

### Use of the stoma as a specimen extraction site

Use of the stoma trephine as an extraction site (where the stoma trephine was used to remove the excised piece of bowel) was an overarching category theme only identified through semistructured interviews and was observed in the non-participant observation:

Extract 1: 'The specimen is handed out. With the specimen removed, I can now see that all that bowel did seem to come out of the stoma hole. Impressive! Won't that have stretched it?'. (Observation NB: BRI0033, end ileostomy, laparoscopic, RDE)

Some surgeons interviewed felt strongly that this would be a high-risk factor for PSH development due to the stretching and extending of the stoma trephine size to accommodate the delivery of the often diseased bowel (see extracts further). Other subthemes included in this overarching category theme were the type of specimen (colon or small bowel) that was brought through the stoma trephine and adjustments to the trephine size to facilitate delivery of the bowel. These factors were also proposed to influence the likelihood of PSH, as shown further:

Extract 2: 'What I feel strongly about is when you use the stoma site to extract a specimen you do predispose them to parastomal hernias so if you're doing a, predictly (sic) if you're doing a subtotal colectomy which is the way it's tempting to do it, you're taking a diseased colon out through a hole which eventually is gonna become an ileostomy site and yes you can narrow it down with sutures afterwards but if we know that parastomal hernia repairs, repair by these sutures have an almost 100% recurrence rate then it's gonna get a parastomal hernia… I think it is important… you're making a hole to get the colon out which is um you know much bigger than the small bowel you're gonna leave there eventually. So technically if you do make the hole too big you're get a parastomal hernia there's no doubt about it'. (HCP: BRI0036, surgeon, lower GI, BRI)

Extract 3: 'You then have to create a very large hole for the specimen to come out of, and, as we discussed before, the size of the hole… Once you've made it big, making it smaller again is really challenging'. (HCP: BRI0023, lower GI, RDE)

Extract 4: 'The other thing is whenever I, if I ever widen the trephine to deliver a specimen, I then to tend to put a couple of stitches in to try and narrow it again, but I don't think that's as good as not opening it in the first place, if that makes sense'. (HCP: BRI0032, surgeon lower GI, RDE)

### Stoma snugness

When the bowel is being brought through the stoma trephine to form the stoma, it may require further adjustments following an assessment of how 'snug' the bowel is within the stoma trephine. a term used by some interview participants to describe the fit of the stoma in the abdominal wall (table 5, extract 1). During the non-participant observations, it was observed that one of the surgeons assessed for 'snugness' by inserting a finger into the stoma trephine and making a further incision to the abdominal wall (table 5, xtract 2). This was also discussed in the

**Table 5** Examples from the qualitative data representing the development of the overarching category theme '*stoma snugness*'

| Subtheme | Extract |
|---|---|
| Assessment of stoma snugness | Extract 1: 'It is better to pull it through and think, "That is a bit snug." Then making a bit of a nick in the posterior sheath to make it a bit wider or using your Langenbeck's to sometimes just stretch it. That does vary between whether you do that sharp or blunt'. (HCP: BRI0022, surgeon, lower GI, RDE) <br> Extract 2: 'With the surgeons index finger he checks for snugness of the trephine. On deciding it is too tight he requests a langenbeck retractor and places it along side his finger, asking the SpR to hold it and retract. Using mackindo scissors he makes a further incision in the peritoneum. Consultant re-checks for snugness on either side of the stoma, checking from inside of the abdomen and outside'. (Observation CM: BRI0034, end ileostomy, converted laparoscopic to open, BRI) <br> Extract 3: 'The traditional has been one finger in, one finger in, and a bit of pulling and tearing and everything else'. (HCP: BRI0023, lower GI, RDE) |
| Mesentery stripping | Extract 4: 'What you're trying to avoid is stripping the mesentery off, cos if you strip the mesentery off the bowel it's ischemic and it will go dusky and flat so you might create a bigger whole and then you've got a risk factor for parastomal hernia'. (HCP: BRI0036, surgeon, lower GI, BRI) <br> Extract 5: 'If the appendices epiploicae are large I'll remove them but I want it under no tension and subcuticular to extramucosal sutures with all the knots buried'. (HCP: BRI0015, surgeon, lower GI, RDE) |

interviews (table 5, extract 3). A further subtheme within this overarching category theme was mesentery stripping, where one surgeon theorised excessive removal of the stoma's mesentery could cause ischaemia and therefore predispose to PSH (table 5, extract 4).

### Closure of other wounds formed during the procedure

Closure of other abdominal wounds formed during the procedure was identified as themes in both the literature and qualitative work. The qualitative work provided rich data on the importance of closing the other abdominal incisions for PSH prevention. For example, BRI00023 felt strongly that restoring the anterior abdominal wall's function through closure of the deep layers using small bite closure with a specific suture size and type was superior for PSH prevention (table 6, extract 1). A further subtheme identified was the order of the wound closure. Some interviewees stated their preference for closing the anterior abdominal wall incisions prior to securing the lumen of the stoma; this practice was also confirmed in the observations (table 6, extracts 3 and 4).

### Creating and securing the stoma lumen

How the stoma lumen was created and secured was an overarching category theme identified in both the literature and qualitative work. Subthemes included efforts to reduce faecal spillage and cleaning of the stoma lumen. Further detail was provided, particularly from non-participant observations, which identified multiple methods of suturing the stoma lumen. It was not clear if

**Table 6** Examples from the qualitative data representing the development of the overarching category theme '*closure of other wounds formed during the procedure*'

| Subtheme | Extract |
|---|---|
| Layers of wound closure (deep layers; Skin layers) | Extract 1: 'How you close the abdominal wall I think is really important, because it then affects how likely the patient is to develop an incisional hernia. If the patient develops an incisional hernia that will impact, because of the mechanics of the abdominal wall on the stoma aperture and then lead to development of parastomal hernias. The two are intimately related. You have to take every possible step to ensure that you have good abdominal wall closure, and restoration of appropriate function, so we tend to use the small bite closure technique using 2–0 PDS delayed absorbable sutures. It's been standard practice now for about two years, particularly for primary surgery'. (HCP: BRI0023, lower GI, RDE) <br> Extract 2: 'Typically I would use a glue to give you a seal. Then you haven't got a dressing extending from the edges of the main wound, that might impinge on where your stoma bag would sit. Also the glue, even if there are leakages, will give you a seal over the centre of the wound'. (HCP: BRI0018, surgeon, hepatobiliary, RDE) |
| Order of wound closure | Extract 3: 'At this point the end of the bowel that I've brought through is typically stapled off, and I will leave it stapled off when it's drawn through the stoma. Then we would finish any further intraabdominal work, close the anterior abdominal wall, close the skin, dress the skin'. (HCP: BRI0018, surgeon, hepatobillary, RDE) <br> Extract 4: '4.0 monocryl on curve and PDS sutures close midline umbilical port site. Mostly out of view with the handheld. Monocryl for port smaller lateral port sites. Wet and dry. Glue used (theory to reduce stoma infections). Two babcocks on distal edges of stapled stoma. Out of view. Lotus dissects the stapled line. Fine tooth forceps and 4.0 monocryl on curve secures stoma to skin with slight spout at 3, 12, 9 then 6 o'clock positions'. (Observation CM: BR0021, end colostomy, laparoscopic, RDE) |

**Table 7** Characteristics of the consensus meeting participants

| Gender | Specialty | Grade | Trust |
|---|---|---|---|
| Male | Colorectal | Consultant | Royal Devon and Exeter |
| Male | Colorectal | Consultant | Queen Elizabeth Hospital |
| Male | Colorectal | Consultant | Bristol Royal Infirmary |
| Male | Colorectal | Consultant | Royal Surrey County Hospital |
| Female | Colorectal | Specialty trainee (ST7) | Southmead Hospital |
| Female | Professor of surgery | Consultant | Bristol Royal Infirmary |
| Male | Professor of health services research | Non-clinician | Bristol Royal Infirmary |
| Male | Health services provider | Non-clinician | Bristol Royal Infirmary |

interview participants perceived these factors to affected PSH rates, but there were many different reported and observed practices to creating and securing the stoma lumen.

### Phase 3: agreement of the final list of data items to be collected in the CRFs
#### Consensus methods
A consensus meeting was held with eight professionals, chaired by the CIPHER study's chief investigator (table 7). Decisions were made through discussion and voting. While consensus on the classification of most data items was simple, others proved more difficult. Some items such as 'length of bowel mobilised' were agreed to be important, but discussion raised concerns over the difficulty in obtaining intraoperative measurements as well as the reliability of surgeons' self-reporting. This item was therefore classified as 'undesirable'. For data items where consensus was unclear, the panel took an open vote by raising their hands. While this method resolved some of the conflict, for a small proportion of the data items, disagreement persisted. These data items were classed as 'desirable' and rediscussed at the end of the meeting. For 15 data items, further discussion did not resolve the differing views, and it was decided that these data items would remain 'desirable' and would be provisionally retained, with a final decision to be made during design of the CRFs.

Four data items that were suggested during the consensus meeting were also included in the final short list: (1) name of the primary procedure, (2) indication for surgery, (3) excision of anterior sheath during trephine formation and (4) excision of posterior sheath during trephine formation. Additional amendments to the list included wording changes, condensing of items and creating of additional response categories. A total of 207 data items were discussed. These items were categorised as essential (n=56), desirable (n=15) and undesirable (n=136). The undesirable items were deemed not important to collect in CRFs and therefore excluded.

Constructing the short list of data items into data fields involved two meetings between the research team, the trials unit staff and the chief investigator of the CIPHER study. Due to the number of items deemed 'essential'

during the consensus meeting, it was decided that the majority of 'desirable' items could not be included in the final CRFs. Each data item was considered individually, and only 3 of the 15 desirable items were included in the final CRFs (use of sutures to buttress the end of the trephine anterior sheath; use of sutures to buttress the end of the trephine posterior sheath; damage to epigastric vessel), because they were simple, and collecting them would not increase the burden on participating surgeons. This decision was supported by the members of the consensus meeting, who were contacted latterly. Amendments were made to phrasing and formatting of the essential items but the content/meaning of the items remained unaltered. The final number of data items included in the CRFs was 59 (see online supplemental file 1, table 4). Combining expert opinion and the trials unit staff knowledge of CRF practicalities was key to ensuring the data items were mapped clearly on to the CRFs, enhancing data collection completeness. These collaborative discussions also enabled any disparity between surgeon and trials unit members expectations and any residual ambiguities between the data items to be resolved.

Adherence to the CRFs and any identified associations between surgical components of stoma formation and the risk of developing parastomal hernia will be reported in the results of the main CIPHER study (see online supplemental file 2, CIPHER case report form extract).

### DISCUSSION
This novel study used a combination of literature reviews, mixed qualitative methods and consensus processes to identify key components of a surgical procedure. This informed development of CRFs for a cohort study investigating risk factors for the development of PSH. A long list containing 207 data items were identified, which was rationalised to 59 to be recorded in the CRFs (figure 2). This application of methods has enabled the development of a systematic process for recording how surgical procedures are conducted in clinical studies, resulting in improvements in transparency of intervention delivery and subsequent implementation in practice. The use of

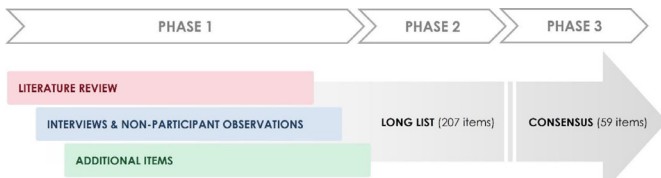

**Figure 2** Overview of study.

scientifically informed CRFs in trials will be valuable to examine protocol adherence and associations between surgical components and outcome.

Previous studies investigating risk factors for the development of PSH have tended to focus on technical variations of interest to the investigators and wider surgical community (eg, prophylactic mesh placement,[16] trephine location[17]). In some, a selection of other technical factors were standardised within the study protocol, although no commentary or justification was provided for these. A typology of surgical procedures, developed to identify components of surgical interventions within RCTs, could have been employed to facilitate the process of ascertaining how much standardisation is required for each intervention component.[18] However, because this approach was designed for RCTs, it does not completely suit non-randomised study designs such as CIPHER. The current study modifies and extends the previously developed methods[18] to incorporate how to develop CRFs to capture the delivery of a surgical intervention within its natural setting, recognising that surgical technique is usually variable.

This is a novel study undertaken to inform the development of CRFs in a detailed and systematic way. Despite this, however, there are potential limitations. Snowballing search strategies are not considered to be as rigorous as systematic database searches of all available literature, tending to favour the identification of well-known literature, and as such, there is a risk that the included papers may not represent the full body of existing literature. As this study focused on achieving saturation of themes, we deemed it unnecessary to identify all the existing evidence and instead were able to identify relevant studies in a more streamlined way. Key to this was the thoughtful selection of the index papers to ensure adequate breath of data and testing the data saturation point by consulting experts and providing a further independent reviewer who coded additional papers to test the end point. The authors particularly acknowledge the limitations of the literature review date (2016). We have reviewed key papers[19 20] published after 2016 that did not yield new themes. Similar to the findings of this study, the authors concluded the lack of consensus on the optimal surgical technique for parastomal hernia prevention to still be present. The methods used to reach consensus can also be criticised. The meeting was chaired by the study Chief Investigator who theoretically could influence responses, and voting was not anonymous. To mitigate against this,

attendees were deliberately selected to ensure a broad range of views were represented.

Inductive qualitative methodological approaches are fundamental to advancing knowledge and insight in a field; these approaches overcome the challenges of becoming constrained by the research team's preconceptions, as is often the limitation of survey methodology. In this project, the qualitative work served this critical role of identifying new items, thus moving the research team beyond the confines of what had already been reported in the literature. It successfully contributed new insights that were incorporated into the final list of items. There is, however, a tension between inductive qualitative methods, principally aimed at generating depth of meaning and understanding, and the intentions underpinning this project: identification of concepts for exhaustive lists. Claims of saturation are difficult in this study, irrespective of how one conceptualises saturation.[13] Unlike qualitative studies that seek saturation of thematic or theoretical constructs (eg, Grounded Theory[21]), our intention was to identify and capture as many examples of 'variation' in surgical practice as possible. We reached a pragmatic decision to cease further interviews during analysis, when themes were noted to regularly repeat, and no new themes or changes to hierarchies were made to the coding framework. This was the point at which further attempts to interview were deemed to be counterproductive, considering the time/funding constraints and the subsequent stages of the study. Including more trusts/individuals in the sample might have illuminated more examples of variation. Nonetheless, this does not undermine or diminish the value of the insights that were contributed by the qualitative work, as would be the case in studies seeking theoretical saturation. New items were added following the consensus meeting, but this was to be expected. The consensus meeting was different in its nature and aims, relative to the observations, interviews and literature reviews. The contributors were explicitly asked to suggest addition of items from the emerging final list and had been exposed to a wide array of data items that will likely have influenced their thoughts and responses. It was anticipated that list items would be added/removed by virtue of the task.

This study identified numerous potential technical factors relating to the future development of PSH, highlighting the complexity of surgical interventions and the need for this to be carefully considered in feasibility or pilot work prior to a main study. Similar to another study conducted by the research team, the research team found the key informants were very eager to participate and to share their expert opinions via observations and interview.[22] Nobody declined to participate, and there was no perceived hesitation from anyone approached. In fact, the surgeons and stoma nurses directly facilitated the observations through introductions to appropriate patients and coordination with the theatre teams. Willingness to participate further validates the feasibility of performing such work. A further key finding from this study is that

unique themes relating to the technical process of stoma formation were identified through both literature and qualitative data collection. Moreover, the broad—and at times conflicting—range of surgical practices/beliefs among participating healthcare professionals may represent a lack of evidence and/or awareness of the evidence. Together, these provide strong rationale for undertaking such work prior to a main study, preferably through multiple methods of data collection. The mixed methods approach has enabled the breadth of data items included in the final CRFs to be more comprehensive than if they were based solely on the opinions and interests of the CIPHER study team. Nesting this work within the early stages of the CIPHER study may have helped to reduce the risk of omitting key variations in stoma formation that may affect PSH development. Other studies may find it helpful to build in a similar phase of feasibility work to provide the time and funding to develop CRFs in such detail. Future work may be needed to streamline the development of CRFs in such detail. On reflection, the research team can recommend performing the literature review while awaiting ethic decisions and local approvals and having multiple researchers performing the literature search in parallel to improve efficiency.

This study used multimodal data collection to enable systematic identification of the components and technical variations of a complex surgical intervention. This informed the rigorous development of CRFs for a future cohort study (CIPHER) aiming to determine risk factors for a key complication of stoma formation surgery (PSH). Through a mixed methods approach, data items were rationalised into a manageable number to include those considered most important. These methods will enable future studies investigating surgical interventions to comprehensively design CRFs to reflect the complexity of the intervention, making it possible to understand exactly how an intervention was delivered within a study so that appropriate conclusions can be drawn.

**Author affiliations**
[1]Bristol NIHR Biomedical Research Centre and Centre for Surgical Research, Population Health Sciences, Bristol Medical School, Bristol University, Bristol, UK
[2]Exeter Surgical Health Services Research Unit (HeSRU), Royal Devon & Exeter Hospital, Exeter, Devon, UK
[3]Academic Department of Surgery, Queen Elizabeth Hospital, University of Birmingham, Birmingham, UK
[4]Department of Colorectal Surgery, University Hospitals Bristol NHS Foundation Trust, Bristol, UK
[5]Department of Oesophago-gastric Surgery, The Royal Surrey County Hospital, Guildford, UK
[6]Department of Colorectal Surgery, Gloucestershire Hospitals NHS Foundation Trust, Cheltenham, UK
[7]Department of Colorectal Surgery, University Hospital of Wales, Cardiff, UK
[8]Department of Surgery, Canisius Wilhelmina Hospital, Nijmegen, Netherlands
[9]Clinical Trials and Evaluation Unit, Bristol Trials Centre, Bristol Medical School, University of Bristol, Bristol, UK

**Acknowledgements** The Cohort study to Investigate the Prevention of parastomal HERnias (CIPHER) team would like to thank those at the Clinical Trials and Evaluation Unit (CTEU) and the Stoma Nurses from Bristol Royal Infirmary and the Royal Devon and Exeter Hospital for their invaluable contribution to this study.

**Contributors** NSB, LR and JB conceived the idea for the study. CM, LR, JB and NSB wrote the protocol. CM undertook the literature and qualitative work supervised by LR, JB and NSB. NS, ID, TP, JS, TR, JB, JT, JR, BR and HB all contributed to the consensus meeting. CM, NSB, LR and JB drafted the manuscript, and all authors read and approved the final manuscript. NB acts as the guarantor for this work.

**Funding** This work was performed as part of the National Institute for Health and Care Research (NIHR) funded UK Cohort study to Investigate the prevention of Parastomal HERnia (CIPHER) Study: Understanding surgery and current practice in stoma formation and developing Patient Reported Outcome Measures for Parastomal Hernia to inform Phase B of CIPHER. Part A was led by JB and supported by the Bristol-based CTEU, Royal College of Surgeons surgical trials centre, the MRC ConDuCT-II (grant number- MR/S001751/1) Hub for Trials Methodology Research and the NIHR Biomedical Research Centre at University Hospitals Bristol and Weston National Health Service (NHS) Foundation Trust and the University of Bristol.

**Disclaimer** The views expressed are those of the author(s) and not necessarily those of the NIHR or the Department of Health and Social Care. JB is an NIHR senior investigator, and NSB is an MRC Clinician Scientist.

**Competing interests** None declared.

**Patient and public involvement** Patients and/or the public were involved in the design, or conduct, or reporting, or dissemination plans of this research. Refer to the Methods section for further details.

**Patient consent for publication** Not applicable.

**Ethics approval** This mixed qualitative study gained NHS Research and Ethics Committee approval from the East Midlands – Nottingham 1 Research Ethics Committee (ref 16/EM/0155) on 1 June 2016. All patients, stoma nurses and surgeons provided written consent to be observed and digitally video recorded in theatre and/or be interviewed. Verbal consent was taken from additional healthcare personnel present in the operating theatre, for observations and digital video recording.

**Provenance and peer review** Not commissioned; externally peer reviewed.

**Data availability statement** Data are available on reasonable request. The datasets generated and analysed during the current study are not publicly available due to confidentiality reasons but are available from the corresponding author on reasonable request.

**ORCID iDs**
Charlotte Murkin http://orcid.org/0000-0002-0838-2694
Jared Torkington http://orcid.org/0000-0002-3218-0574
Barnaby Reeves http://orcid.org/0000-0002-5101-9487
Jane Blazeby http://orcid.org/0000-0002-3354-3330
Natalie S Blencowe http://orcid.org/0000-0002-6111-2175

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
