## [Reviewer comments · BMJ Open]

ARTICLE DETAILS

TITLE (PROVISIONAL)	What should be included in case report forms? Development and application of novel methods to inform surgical study design: a mixed methods case study in parastomal hernia prevention
AUTHORS	Murkin, Charlotte; Rooshenas, Leila; Smart, Neil; Daniels, IR; Pinkney, Tom; Shabbir, J; Rockall, Timothy; Bennett, J; Torkington, Jared; Randall, J; Brandsma, HT; Reeves, Barnaby; Blazeby, Jane; Blencowe, Natalie

VERSION 1 – REVIEW

REVIEWER	Weyhe, Dirk Pius-Hospital Oldenburg, University Hospital for Visceral Surgery
REVIEW RETURNED	14-Mar-2022

GENERAL COMMENTS	Dear authors, congratulations on tackling this very important issue of the creation of standardized, evidence based CRFs. It is in my opinion long overdue to make this step. The way you established your CRF was very well described and might be a help to future scientists wanting to implement an CRF for their purposes. One could argue about some points, like the way the consensus meeting was held, or your way of literature search. However, as I see it, you choose a very pragmatic approach with only negligible limitations. Therefore, I do not have any major comments with regards to this manuscript. However, I would recommend to include some statement on why you believe that you do not stay in some kind of research bubble of authors just citing themselves or close associates with this snowball approach. At least in my experience, researchers tend to cite each other and sometimes forget to look to much beyond their known group of associates. Or papers with opinions the authors do not agree with, are not cited as much. This might create some kind of bubble of papers only citing each other, analogous to certain social media circles, nowadays. Judging by the cited papers and from my knowledge of the current literature, this might not have been an issue here. However, in my opinion it might have been better to start with at least two or three index papers with only little overlap in their reference lists, to circumvent this potential problem. Could you please comment on that whole issue, and include a statement to that regard in the manuscript? With kind regards
---

REVIEWER	MADRAZO GONZALEZ, ZOILO Bellvitge University Hospital, General and Digestive Surgery
REVIEW RETURNED	24-May-2022

GENERAL COMMENTS	I congratulate you for the meticulous work done.
--

REVIEWER	Krogsgaard, Marianne Zealand University Hospital Koge, Dept Surgery
-----------------	--

REVIEW RETURNED	29-Jun-2022
-------------

GENERAL COMMENTS	Thank you for the opportunity to review this manuscript. Overall, the extensive work is a strength of the study. My comments relate mainly to the Methods section and the need for clarity and references to relevant literature. Introduction Clear Methods Overall, the methods section, although comprising multiple descriptions, is well written and comprehensible. However, the section lack references to relevant literature on included methods, e.g., ‘snowballing’ (systematic literature review), what method was used for ‘content analysis’ (qualitative study), ‘saturation’, ‘non-participant observation (qualitative mixed methods) and so on. The methods section would also profit from addressing the benefits of triangulation. Saturation - please provide the definition used in the study (saturation is debated and there are different viewpoints on saturation in the literature) and include relevant references. Operations/participants were purposefully sampled (p.7, line 47) – please state more clearly, how this was done and what variation was intended. Did the researchers know the interviewees? Did observations and semi-structured interviews follow a guide/template? How were they developed? The structure of the themes/categories is somewhat difficult to tease out from the examples given in Supplementary Tables 1, 2 and 3. It would be helpful if the ‘themes’ and ‘overarching categories’ shown in the Supplementary Tables aligned with the content provided in Figure 1. Patient and public involvement (p.9, line 57). Please provide some more information on the involvement, e.g., incorporated in the suggested Figure X, see below ‘Tables and Figures’. Results Literature review: please state the publication year for the first and latest published article reviewed. It seems that literature no later than 2016 was included. Please check that the numbers given in Table 1 correspond with the text on page 10, line 13. How many health professionals were eligible for inclusion for interviews/observations? And how many consented/declined? Phases 2 and 3. It is somewhat surprising that 12 new items were added to the long list by two senior colorectal surgeons, p. 13, line 52. Likewise, four data items were added during the consensus meeting (p. 23, line 3ff). Could this indicate that saturation was not achieved or that the sampling strategy was not ideal? How do the authors explain this? Please provide some reflections on this issue in the discussion section/limitations. Discussion
--

	Saturation – please include reflections as pointed to above, phases 2 and 3. Could the inclusion of more recent literature have provided new items? Please reflect on this in the discussion/limitation section. How did health care professionals react to the non-participant observations and video recordings? Please reflect on, e.g., the willingness to participate, as this information may be important in the design of future studies. The study is described as time-consuming in the Article summary p.4, line 21. It would add value to the reader if the authors also address this issue in the discussion section. Is it possible to estimate the timeframe for this work (years/months) and the efforts put into the study? What is the future perspective for the use of this method – do the authors have suggestions on how to bring down the time? Please provide the reader with some of the authors' considerations/reflections. Tables and figures I suggest the inclusion of a 'Figure X' showing the different methods for data collection and Phase 1-3. This could provide the reader with a nice overview of the study.
--	--

VERSION 1 – AUTHOR RESPONSE

Reviewer 1 comments

1. Congratulations on tackling this very important issue of the creation of standardized, evidence based CRFs. It is in my opinion long overdue to make this step. The way you established your CRF was very well described and might be a help to future scientists wanting to implement an CRF for their purposes. One could argue about some points, like the way the consensus meeting was held, or your way of literature search. However, as I see it, you choose a very pragmatic approach with only negligible limitations. Therefore, I do not have any major comments with regards to this manuscript.

Reply: We thank the reviewer for their support.

2. However, I would recommend to include some statement on why you believe that you do not stay in some kind of research bubble of authors just citing themselves or close associates with this snowball approach. At least in my experience, researchers tend to cite each other and sometimes forget to look to much beyond their known group of associates. Or papers with opinions the authors do not agree with, are not cited as much. This might create some kind of bubble of papers only citing each other, analogous to certain social media circles, nowadays. Judging by the cited papers and from my knowledge of the current literature, this might not have been an issue here. However, in my opinion it might have been better to start with at least two or three index papers with only little overlap in their reference lists, to circumvent this potential problem. Could you please comment on that whole issue, and include a statement to that regard in the manuscript?

Reply: We completely agree with the reviewer that good index paper selection is key to ensuring the breath and richness of data. We also agree that a snowballing strategy favours the identification of well-known literature, risking an underrepresentation of the full body of existing literature. We have provided an expanded explanation in the Methods section and also discussed this limitation in detail within the Discussion.

Revision: The following section in the Methods (page 6, line 21 onwards) now reads:

Snowballing began using an index paper (2), defined as a high-quality paper known to contain information relevant to the study question. The index paper (3) was selected by two senior surgeons both with extensive experience of surgical research. Papers cited by the index paper were then

screened (backward snowballing) using broad inclusion criteria (i.e. written in English and providing technical details of small or large bowel stoma formation). Once backward snowballing was complete, all papers citing the index paper were retrieved (forward snowballing) using a citation tracking service. A further five index papers (3–7) were then selected to continue snowballing. These papers were identified through discussion with experts, with a focus on data-rich, high-quality literature (e.g. reviews). At the end of the snowballing process, two newly published RCTs were identified. Because of their potential relevance, it was agreed they should form part of the literature review but further snowballing was not undertaken (8-9).

We have also added information about snowballing to the limitations section of the Discussion (page 19, line 14 onwards):

This is a novel study undertaken to inform the development of CRFs in a detailed and systematic way. Despite this, however, there are potential limitations. Snowballing search strategies are not considered to be as rigorous as systematic database searches of all available literature, tending to favour the identification of well-known literature, and as such there is a risk that the included papers may not represent the full body of existing literature. As this study focused on achieving saturation of themes, we deemed it unnecessary to identify all the existing evidence and instead were able to identify relevant studies in a more streamlined way. Key to this was the thoughtful selection of the index papers to ensure adequate breadth of data and testing the data saturation point by consulting experts and providing a further independent reviewer who coded additional papers to test the end point.

Reviewer 2 comments

I congratulate you for the meticulous work done.

Reply: Again, we thank the reviewer for their support.

Reviewer 3 comments

Thank you for the opportunity to review this manuscript. Overall, the extensive work is a strength of the study. My comments relate mainly to the Methods section and the need for clarity and references to relevant literature.

Reply: We thank Reviewer 3 for their support.

Methods:

1. Overall, the methods section, although comprising multiple descriptions, is well written and comprehensible. However, the section lacks references to relevant literature on included methods, e.g., 'snowballing' (systematic literature review), what method was used for 'content analysis' (qualitative study), 'saturation', 'non-participant observation (qualitative mixed methods) and so on.

Reply: We apologise for omitting these references.

Revisions: We have added the following references to the appropriate places in the text (references pertaining to 'saturation' are dealt with separately below):

Non-participant observation, triangulation, and snowballing – Richie J. and Lewis J. (2003) *Qualitative Research Practice: A Guide for Social Science Students and Researchers*. Sage Publications, London.

Content analysis and topic guide - Bryman A. (2016) *Social Research Methods*. Oxford University Press.

2. The methods section would also profit from addressing the benefits of triangulation.

Reply: We have now included an explanation that triangulation allows for a more comprehensive understanding of the phenomenon being investigated, with examples and a reference from Richie J.

and Lewis J. (2003) *Qualitative Research Practice: A Guide for Social Science Students and Researchers*. Sage Publications, London.

Revisions:

The following section in the Methods (page 7, line 15 onwards) now reads:

This work had three objectives, to i) confirm, refine, refute and develop the 'known and reported' factors identified in the literature work, ii) uncover and explore 'unknown but practised' technical variations in stoma formation that may influence PSH development, and iii) triangulate the non-participant observation and semi-structured interview to allow for a more comprehensive understanding of stoma formation techniques (2). For example, interviews allowed for opportunity to explore and explain the practices observed, and tease out practices that informants reported conducting outside of the observed cases; observations allowed for direct insight into real cases, providing detail that might have been omitted from interviews.

4. Saturation - please provide the definition used in the study (saturation is debated and there are different viewpoints on saturation in the literature) and include relevant references.

Reply: We apologise for omitting this definition from the manuscript. We have now provided further information about the issue of saturation in both the Methods and Discussion sections. We have included a reference from Grady MP. *Qualitative and Action Research: A Practitioner Handbook*. Phi Delta Kappa International; 1998. 68 p.

Revisions:

The following section in the Methods (page 8, line 6 onwards) now reads:

Sampling continued up until the point of saturation, whereby during parallel analysis no new themes were uncovered and additional data were not adding any further explanatory or descriptive benefit. The decision about when to cease further sampling/data collection was guided by the core research team's (CM, LR, NB) assessment of the 'tipping point', whereby further interviews were not deemed unlikely to be of global benefit for fulfilling the study aim and objectives. This was similar to what others have referred to as the point of 'informational redundancy' (13): a point at which "new data tend to be redundant of data already collected" (14). In this case, we reached a point whereby no new items were emerging from concurrent analysis of interviews. While there was a possibility of this changing with further data collection, the team made a judgment to cease, once the benefits of continued sampling were weighed up against the consequences of delaying the next steps of the mixed-methods project.

We have also amended the Discussion (page 19, line 32 onwards) to now read:

Inductive qualitative methodological approaches are fundamental to advancing knowledge and insight in a field; these approaches overcome the challenges of becoming constrained by the research team's preconceptions, as is often the limitation of survey methodology. In this project, the qualitative work served this critical role of identifying new items, thus moving the research team beyond the confines of what had already been reported in the literature. It successfully contributed new insights which were incorporated into the final list of items. There is, however, a tension between inductive qualitative methods, principally aimed at generating depth of meaning and understanding, and the intentions underpinning this project: identification of concepts for exhaustive lists. Claims of saturation are difficult in this study, irrespective of how one conceptualises saturation (13). Unlike qualitative studies that seek saturation of thematic or theoretical constructs (e.g. grounded theory (19), our intention was to identify and capture as many examples of 'variation' in surgical practice as possible. We reached a pragmatic decision to cease further interviews during analysis, when themes were noted to regularly repeat and no new themes or changes to hierarchies were made to the coding framework. This was the point at which further attempts to interview were deemed to be counter-productive, considering the time/funding constraints, and the subsequent stages of the study. Including more trusts/individuals in the sample might have illuminated more examples of variation.

Nonetheless, this does not undermine or diminish the value of the insights that were contributed by the qualitative work, as would be the case in studies seeking theoretical saturation. New items were added following the consensus meeting, but this was to be expected. The consensus meeting was different in its nature and aims, relative to the observations, interviews, and literature reviews. The contributors were explicitly asked to suggest addition of items from the emerging final list, and had been exposed to a wide array of data items that will likely have influenced their thoughts and responses. It was anticipated that list items would be added/removed, by virtue of the task.

4. Operations/participants were purposefully sampled (p.7, line 47) – please state more clearly how this was done and what variation was intended.

Reply: When identifying surgeons to participate we intended to include surgeons from a variety of specialties and grades. In practice, however, the majority of surgeons who form stomas in the planned setting are colorectal specialists. Colorectal surgeons also lead the management of patients with PSH and are therefore most likely to be familiar with existing and emerging literature in this area, including PSH prevention. Upper gastro-intestinal surgeons also occasionally form stomas when providing out-of-hours emergency care. Although these surgeons would not be considered experts, we felt it would be important to try and recruit them, as it is possible they use different methods to create stomas. We therefore adopted a 'key informant' approach. We have included a reference from Marshall MN. Sampling for qualitative research. Family practice. 1996;13(6):522-6.)

Revision:

The following section in the Methods (page 7, line 30 onwards) now reads:

Operations featuring stoma formation were purposefully sampled from two NHS hospital centres, with the aim of observing stoma formation performed by different surgeons, rather than the same surgeon creating multiple stomas. The sampling of healthcare professionals was based on a 'key informant' purposeful approach, whereby individuals with exclusive specialist expertise are selected as the most productive sample to speak to and observe (12). Clinicians with a role in conducting stoma surgery or providing aftercare were therefore approached. This included stoma nurses and surgeons performing stoma surgery either routinely (i.e. colorectal surgeons) or as part of the emergency service (i.e. Upper GI and colorectal surgeons), at both consultant and registrar level. Approximately 32 surgeons (including registrars) were performing stoma formation and six stoma nurses were employed at the time of data collection between the two trusts. Sampling continued up until the point of saturation, whereby during parallel analysis no new themes were uncovered and additional data were not adding any further explanatory or descriptive benefit.

5. Did the researchers know the interviewees?

Reply: CM had previously met three of the surgeons. The remaining surgeons and stoma nurses were unknown to CM.

Revisions:

The Results section has now been amended (page 11, line 6) to read:

Six procedures involving stoma formation were observed (Table 2). The ages of patients whose stoma formation was observed ranged from 31-84 years. Thirteen health professionals were interviewed across two centres (Table 3). No surgeon that was approached declined an interview. However, it was not possible to interview four willing surgeons due to their conflicting time commitments. Six stoma nurses were eligible for inclusion, three consented and none further were approached. Interviews were conducted by CM who was acquainted with three of the surgeons interviewed. Interviews lasted between 17:22 and 1:13:95 minutes (mean = 27:45 minutes). There was a broad range of experience across the clinicians interviewed and consensus meeting participants: their length of time in role ranged from <1-16 and <1-17 years, respectively.

6. Did observations and semi-structured interviews follow a guide/template? How were they

developed?

Reply: We apologise that this information was missing from the manuscript. Additional text has now been provided and a reference to Bryman A. Social Research Methods. Oxford University Press; 2016. 785 p. has been added.

Revision:

We have amended the Methods section (page 8, line 25) to now read:

Semi-structured interviews were conducted by CM using a topic guide. The topic guide was a series of open-ended questions and prompts/probes, arranged into broad topics to guide the interviewer (11). Informed by literature review findings, the topic guide was pre-tested in two interviews (not included in the analysis) to test for flow, relevance and whether the questions provoked discussion. Interview topics derived from the category themes identified during the literature review and were updated iteratively as new themes emerged during the interviews.

7. The structure of the themes/categories is somewhat difficult to tease out from the examples given in Supplementary Tables 1, 2 and 3. It would be helpful if the 'themes' and 'overarching categories' shown in the Supplementary Tables aligned with the content provided in Figure 1.

Reply: Thank you for this helpful suggestion. Figure 1 provides an example of the coding framework developed in the literature review for the overarching theme of 'mesh'. We have amended two of the Tables so they encompass the same overarching theme as Figure 1 (mesh) We hope that using a common theme helps the reader. We have not amended the other table in the Appendix as this contains important information about Trepine formation that complements the article text, which would be lost.

Revision:

We have amended Additional file 1, Appendices, Tables 1 and 3 to contain details about the theme 'mesh', to match Figure 1.

8. Patient and public involvement (p.9, line 57). Please provide some more information on the involvement, e.g., incorporated in the suggested Figure X, see below 'Tables and Figures'.

Reply: Once again we apologise for this omission. PPI work was a central feature of the CIPHER study and we have now added this information to the body of the paper.

Revisions:

Underneath the 'Patient and public involvement (PPI)' heading (Methods, page 10, page 21), we have added an additional heading 'Patient and public involvement' which reads:

PPI work was central to the design and conduct of the CIPHER study. During conception of the project, it was discussed at the Association of Coloproctology of Great Britain and Ireland Patient Consultation Exercise (attended by representatives from national patient support groups in) and the study team met separately with patient representatives from colostomy, ileostomy and urostomy organisations. Two patients are members of the Study Steering Committee. Items relating to the seniority of surgeon forming the stoma, and whether they were supervised, were of particular interest to patients attending the PPI meeting and they agreed strongly that these should be included in the CRFs.

Results:

1. Literature review: please state the publication year for the first and latest published article reviewed.

It seems that literature no later than 2016 was included. Please check that the numbers given in Table 1 correspond with the text on page 10, line 13.

Reply: Many thanks for bringing this to our attention. We have revisited our literature tracker and recognised an error, many thanks for bringing this to our attention. The reviewer is correct that no literature beyond 2016 was included – we have responded to this in point Phases 2 and 3: Question 3 below.

Revision:

The Results section now reads (page 10, line 30):

A total of 490 480 references were screened and of these, 130 articles finally included, with publication dates ranging from 1958 – 2016.

2. How many health professionals were eligible for inclusion for interviews/observations? And how many consented/declined?

Reply: It is difficult to be completely accurate with the number of healthcare professionals eligible for inclusion, given that trainees frequently rotate hospitals (and often at different times). We estimate the number of eligible surgeons to be 30 including trainees.

Revision:

The Methods section (page 8, line 4) has been updated to read as:

Approximately 32 surgeons (including registrars) were performing stoma formation and six stoma nurses were employed at the time of data collection between the two trusts.

The Results (page 11, line 6 onwards) have been amended to read:

Six procedures involving stoma formation were observed (Table 2). The ages of patients whose stoma formation was observed ranged from 31-84 years. Thirteen health professionals were interviewed across two centres (Table 3). No surgeon that was approached declined an interview. However, it was not possible to interview four willing surgeons due to their conflicting time commitments. Six stoma nurses were eligible for inclusion, three consented and none further were approached. Interviews were conducted by CM who was acquainted with three of the surgeons interviewed. Interviews lasted between 17:22 and 1:13:95 minutes (mean = 27:45 minutes. There was a broad range of experience across the clinicians interviewed and consensus meeting participants: their length of time in role ranged from <1-16 and <1-17 years, respectively.).

Phases 2 and 3:

1. It is somewhat surprising that 12 new items were added to the long list by two senior colorectal surgeons, p. 13, line 52. Likewise, four data items were added during the consensus meeting (p. 23, line 3ff). Could this indicate that saturation was not achieved or that the sampling strategy was not ideal? How do the authors explain this? Please provide some reflections on this issue in the discussion section/limitations.

Reply: Thank you for these very helpful comments. We fully agree, and have now added more detail to the manuscript to provide a more transparent account of how we made judgments about when to stop interviews. We have also now reflected on different conceptualisations of saturation, and the possibility that we might have identified more variations had we continued interviews – particularly if we had the resources to conduct interviews in more sites. We have also added some reflections on the addition of items arising from the consensus meeting, to draw attention to the fact that consensus

meeting contributors' involvement was quite different to that of interview participants, and that there was an expectation that the long list of items generated from the interviews, observations, and literature would be edited (deletions and additions).

Revision:

The following section in Methods (page 8, line 8 onwards) now reads:

The decision about when to cease further sampling/data collection was guided by the core research team's (CM, LR, NB) assessment of the 'tipping point', whereby further interviews were not deemed unlikely to be of global benefit for fulfilling the study aim and objectives. This was similar to what others have referred to as the point of 'informational redundancy' (13): a point at which "new data tend to be redundant of data already collected" (14). In this case, we reached a point whereby no new items were emerging from concurrent analysis of interviews. While there was a possibility of this changing with further data collection, the team made a judgment to cease, once the benefits of continued sampling were weighed up against the consequences of delaying the next steps of the mixed-methods project.

The Discussion (page 19, line 30 onwards) has also been amended and now reads:

Inductive qualitative methodological approaches are fundamental to advancing knowledge and insight in a field; these approaches overcome the challenges of becoming constrained by the research team's preconceptions, as is often the limitation of survey methodology. In this project, the qualitative work served this critical role of identifying new items, thus moving the research team beyond the confines of what had already been reported in the literature. It successfully contributed new insights which were incorporated into the final list of items. There is, however, a tension between inductive qualitative methods, principally aimed at generating depth of meaning and understanding, and the intentions underpinning this project: identification of concepts for exhaustive lists. Claims of saturation are difficult in this study, irrespective of how one conceptualises saturation (13). Unlike qualitative studies that seek saturation of thematic or theoretical constructs (e.g. Grounded Theory (21)), our intention was to identify and capture as many examples of 'variation' in surgical practice as possible. We reached a pragmatic decision to cease further interviews during analysis, when themes were noted to regularly repeat and no new themes or changes to hierarchies were made to the coding framework. This was the point at which further attempts to interview were deemed to be counter-productive, considering the time/funding constraints, and the subsequent stages of the study. Including more trusts/individuals in the sample might have illuminated more examples of variation. Nonetheless, this does not undermine or diminish the value of the insights that were contributed by the qualitative work, as would be the case in studies seeking theoretical saturation. New items were added following the consensus meeting, but this was to be expected. The consensus meeting was different in its nature and aims, relative to the observations, interviews, and literature reviews. The contributors were explicitly asked to do a very different task to our interviewees, who were solely expected to reflect on their practice. Consensus meeting contributors were explicitly asked to suggest addition of items from the emerging final list, and had been exposed to a wide array of data items that will likely have influenced their thoughts and responses. It was anticipated that list items would be added/removed, by virtue of the task.

2. Saturation – please include reflections as pointed to above, phases 2 and 3.

Reply: We have now addressed this issue in the points above (In Methods, question 4).

3. Could the inclusion of more recent literature have provided new items? Please reflect on this in the discussion/limitation section.

Reply: Thank you for highlighting this. We agree this could be a limitation to this study and we have considered this when responding to your earlier point.

Revision:

The Discussion section (page 19, line 23 onwards) now reads:

The authors particularly acknowledge the limitations of the literature review date (2016). We have reviewed key papers (19, 20) published after 2016 which did not yield new themes. Similar to the findings of this study, the authors concluded the lack of consensus on the optimal surgical technique for parastomal hernia prevention to still be present.

3. How did health care professionals react to the non-participant observations and video recordings? Please reflect on, e.g., the willingness to participate, as this information may be important in the design of future studies.

Reply: Thank you for drawing our attention to this interaction which is important to report on. Our experience from other studies is that surgeons may initially express hesitation, yet after a short time 'forget' they are being observed/videoed).

Revision:

The Discussion section (page 20, line 20 onwards) has been amended and now reads:

This study identified numerous potential technical factors relating to the future development of PSH, highlighting the complexity of surgical interventions and the need for this to be carefully considered in feasibility or pilot work prior to a main study. Similar to another study conducted by the research team, the research team found the key informants were very eager to participate and to share their expert opinions, via observations and interviews (22). Nobody declined to participate and there was no perceived hesitation from anyone approached. In fact, the surgeons and stoma nurses directly facilitated the observations through introductions to appropriate patients and co-ordination with the theatre teams. Willingness to participate further validates the feasibility of performing such work.

5. The study is described as time-consuming in the Article summary p.4, line 21. It would add value to the reader if the authors also address this issue in the discussion section. Is it possible to estimate the timeframe for this work (years/months) and the efforts put into the study? What is the future perspective for the use of this method – do the authors have suggestions on how to bring down the time? Please provide the reader with some of the authors' considerations/reflections.

Reply: Thank you for highlighting this as a consideration.

Revision:

The Discussion section (page 21, line 3 onwards) has been amended and now reads:

. Other studies may find it helpful to build in a similar phase of feasibility work to provide the time and funding to develop CRFs in such detail. Future work may be needed to streamline the development of CRFs in such detail. On reflection the research team can recommend performing the literature review while awaiting ethic decisions and local approvals and having multiple researchers performing the literature search in parallel to improve efficiency.

Tables and figures:

1. I suggest the inclusion of a 'Figure X' showing the different methods for data collection and Phase 1-3. This could provide the reader with a nice overview of the study.

Reply: We thank the reviewer for this suggestion and have included a new Figure within the updated manuscript.

Revisions: The new Figure is provided below.

(Figure 2: Overview of study)

We now hope that you will find the revised manuscript appropriate for publication in BMJ Open.

VERSION 2 – REVIEW

REVIEWER	Krogsgaard, Marianne Zealand University Hospital Koge, Dept Surgery
REVIEW RETURNED	01-Sep-2022
GENERAL COMMENTS	I would like to congratulate the authors with the manuscript. I believe the changes to the manuscript, can ease application of such a novel approach in future studies.